# Evaluation of the Oral Bacterial Genome and Metabolites in Patients with Wolfram Syndrome

**DOI:** 10.3390/ijms24065596

**Published:** 2023-03-15

**Authors:** E. Zmysłowska-Polakowska, T. Płoszaj, S. Skoczylas, P. Mojsak, M. Ciborowski, A. Kretowski, M. Lukomska-Szymanska, A. Szadkowska, W. Mlynarski, A. Zmysłowska

**Affiliations:** 1Department of Endodontics, Medical University of Lodz, 92-213 Lodz, Poland; 2Department of Clinical Genetics, Medical University of Lodz, 92-213 Lodz, Poland; 3Clinical Research Centre, Medical University of Bialystok, 15-276 Bialystok, Poland; 4Department of Endocrinology, Diabetology and Internal Medicine, Medical University of Bialystok, 15-276 Bialystok, Poland; 5Department of General Dentistry, Medical University of Lodz, 92-213 Lodz, Poland; 6Department of Pediatrics, Diabetology, Endocrinology and Nephrology, Medical University of Lodz, 92-213 Lodz, Poland; 7Department of Pediatrics, Oncology and Hematology, Medical University of Lodz, 92-213 Lodz, Poland

**Keywords:** Wolfram syndrome, neurodegeneration, gingival samples, oral microbiome, metabolomics

## Abstract

In Wolfram syndrome (WFS), due to the loss of wolframin function, there is increased ER stress and, as a result, progressive neurodegenerative disorders, accompanied by insulin-dependent diabetes. The aim of the study was to evaluate the oral microbiome and metabolome in WFS patients compared with patients with type 1 diabetes mellitus (T1DM) and controls. The buccal and gingival samples were collected from 12 WFS patients, 29 HbA1c-matched T1DM patients (*p* = 0.23), and 17 healthy individuals matched by age (*p* = 0.09) and gender (*p* = 0.91). The abundance of oral microbiota components was obtained by Illumina sequencing the 16S rRNA gene, and metabolite levels were measured by gas chromatography–mass spectrometry. *Streptococcus* (22.2%), *Veillonella* (12.1%), and *Haemophilus* (10.8%) were the most common bacteria in the WFS patients, while comparisons between groups showed significantly higher abundance of *Olsenella*, *Dialister*, *Staphylococcus*, *Campylobacter*, and *Actinomyces* in the WFS group (*p* < 0.001). An ROC curve (AUC = 0.861) was constructed for the three metabolites that best discriminated WFS from T1DM and controls (acetic acid, benzoic acid, and lactic acid). Selected oral microorganisms and metabolites that distinguish WFS patients from T1DM patients and healthy individuals may suggest their possible role in modulating neurodegeneration and serve as potential biomarkers and indicators of future therapeutic strategies.

## 1. Introduction

Wolfram syndrome (WFS; DIDMOAD syndrome, OMIM#222300) is a rare (1:770,000) autosomal recessively inherited disease with the presence of pathogenic variants in the *WFS1* gene, in which the life expectancy of patients is approximately 30–40 years. The criteria for the clinical diagnosis of WFS are the coexistence of insulin-dependent diabetes mellitus with optic nerve atrophy accompanied by diabetes insipidus and deafness. Other symptoms include neurological, urodynamic, and endocrine disorders [1,2,3]. In WFS the clinical manifestation of diabetes mellitus occurs early, usually at the age of 4–7 years and is the first symptom of the syndrome. It is a non-autoimmune form of insulin-dependent diabetes mellitus resulting from selective pancreatic β-cells loss and impaired insulin secretion. However, the essence of Wolfram syndrome is the progressive neurodegeneration accompanying diabetes, which leads to multiple disorders and, consequently, to the premature death of the patient [4,5]. WFS is characterized by a deficiency of wolframin located in the membrane of the endoplasmic reticulum (ER), whose loss of function leads to an increase in ER stress and apoptosis of many target cells. This results in clinical events present in patients including progressive visual impairment based on optic atrophy, progressive hearing impairment, neurogenic bladder, and neurological disorders [1,6]. Interestingly, despite the well-known and documented genetic background of WFS, there are indications that epigenetic and environmental factors also influence the clinical manifestation and course of the syndrome. It is worth mentioning the different phenotype and course of the disease observed within the same families and in patients carrying the same causative variants [7], which may suggest the influence of different infections or pathogens. Moreover, new research indicates that WFS patients show a unique metabolic fingerprinting in serum, distinguishing them from patients with type 1 diabetes mellitus (T1DM) and serum levels of sphinganine derivative appear to be a marker of ongoing neurodegeneration in WFS patients [8].

Furthermore, recent reports on Alzheimer’s disease (AD)—as a model neurodegenerative disease—highlight the role of periodontal status and the involvement of specific bacteria in the induction and progression of neurodegeneration. These reports indicate that once the protective barrier of the host’s oral tissues is broken, bacteria migrate and contribute in different ways to the pathologies observed in AD [9,10]. On the other hand, it is well known that periodontal disease is a risk factor for cardiovascular disease, diabetes, and other chronic diseases [11]. Moreover, new findings have shown interactions between specific bacterial species and genes identified in genome-wide association studies (GWAS) as responsible for the development of these diseases. Thus, these data suggest important genetic-environmental interactions between the presence of oral bacteria and a specific genetic predisposition to neurodegeneration or changes in gene expression in conditions where periodontal disease is a contributing factor [12,13]. Furthermore, the identification by GWAS of polymorphisms associated with various environmental factors may, as in AD, encourage the development of diagnostic and therapeutic strategies [14].

The inadequacy of microbiological methods in the identification of different types of bacteria and the increasing availability of new methods for the detection of the bacterial genome have forced attempts to make wider use of the latter. In recent years, due to the development of the latest high-throughput sequencing techniques such as next-generation sequencing (NGS), bacterial metagenomic studies have become cheaper and more accessible. Very often, classical microbiological tests can give false-negative results because some microorganisms may not be able to grow under given laboratory conditions and/or in the presence of other bacterial species. These methodological difficulties can be eliminated with bacterial genome studies. As a result, most of the currently known bacterial species can be characterized both quantitatively and qualitatively [15,16,17,18].

The aim of this study was to evaluate the bacterial genome in gingival and buccal fluid samples of patients with Wolfram syndrome in comparison with patients with type 1 diabetes without neurodegeneration and healthy subjects, with subsequent metabolomic analysis of the gingival crevicular fluid (GCF).

## 2. Results

A total of 5,221,745 reads attributed to bacteria were obtained. The average number of reads assigned to a bacterial type per sample was 82884. On average, 12.8% of reads per sample were not assigned to any bacterial type. Among the bacterial genera identified by the NGS method, *Streptococcus* (22.2%), *Veillonella* (12.1%), and *Haemophilus* (10.8%) were the most common in the study group of the WFS patients, followed by *Prevotella* (8.6%) and *Neisseria* (1.8%), while *Streptococcus* was the most common in the control group, accounting for up to 39.3%. The following types occurred in the control group: *Haemophilus* (18.1%), *Prevotella* (5%), and *Neisseria* and *Veilonella* (4.1% each, respectively). In the comparison group of patients with T1DM, the most commonly identified bacterial type was *Streptococcus* (30.8%), *Haemophilus* (10.3%), and *Veilonella* (8%), followed by *Neisseria* (6.8%) and *Prevotella* (5%) (Figure 1). 

Figure 2 shows the complete identification of the bacterial genera present in all study groups, which enabled their clustering (Figure 2).

The overall core microbiome analysis is presented in Appendix A. Samples from the study, comparison, and control groups differed significantly in all five measures of alpha diversity analysis used: Chaol (*p* = 0.0199), ACE (*p* = 0.0423), Simpson (*p* < 0.0001), Fisher (*p* = 0.0004), and Shannon (*p* < 0.0001) (Figure 3, Appendix A).

An analysis of beta diversity also indicated a statistically significant difference between the study, comparison, and control groups (*p* < 0.001) (Figure 4).

Next, the most important bacteria identified in the microbiome data analysis were assessed using a univariate analysis method (Appendix A). Comparisons were made between groups for the five most statistically significant bacterial genera identified: *Olsenella*, *Dialister*, *Staphylococcus*, *Campylobacter*, and *Actinomyces*, indicating a significantly higher abundance value in the WFS study group compared with the control and comparison groups (Figure 5A–E) (*p* < 0.001; *p* < 0.001; *p* < 0.001; *p* < 0.001; and *p* < 0.001, respectively).

Moreover, after processing the GC-MS data, 120 metabolic signals were detected for all comparisons. Signal clustering and filtering processes yielded a total of 56 metabolites, of which a group of 25 metabolites could be annotated. CV values (for QC samples) were calculated. Based on the results, it was confirmed that the CV values for 13 metabolites were below 30% (Table 1).

Metabolite intensities were then compared between the studied groups. Consequently, eight metabolites (*p*-value <0.05 * and <0.01 **) in all comparisons could be distinguished and significant metabolites of GCF samples are shown in Figure 6. These metabolites mainly belong to carboxylic acids, amino acids, and carbohydrates. 

Statistically significant differences were observed in the intensity of six metabolites: acetic acid (FC = 2.3), lactic acid (FC = 3.0), valine (FC = 3.5), benzoic acid (FC = 1.3), glycerol (FC = 5.8), and succinic acid (FC = 3.4). This finding can distinguish WFS patients from healthy subjects, while four metabolites (acetic acid, propionic acid, lactic acid, and benzoic acid) can differentiate WFS from T1DM patients. The intensity of such metabolites as acetic acid (FC = 2.1), lactic acid (FC = 2.5), and benzoic acid (FC = 1.4) were significantly higher in WFS patients compared with both T1DM patients and healthy subjects (Figure 6).

In addition, to assess the potential of significant metabolites as WFS predictors, ROC curves were constructed using the relative metabolite contents of the studied groups (Figure 7A). Multivariate (AUC = 0.861) (Figure 7B) and individual ROC curves (Figure 7C) were constructed for the three metabolites showing the best discriminatory power (acetic acid, benzoic acid, and lactic acid).

## 3. Discussion

For the first time, the bacterial genome from the gingival and buccal fluid of patients with WFS syndrome was evaluated and compared with results from patients with isolated insulin-dependent diabetes and healthy individuals. Moreover, the metabolites present in the GCF of WFS patients were also evaluated and compared with a group of T1DM patients and healthy subjects. The present results confirmed that the most frequently represented oral bacteria in WFS patients were those of the *Streptococcus*, *Haemophilus*, and *Veillonella* genera. In addition, a significant diversity of bacterial types was observed both within the study groups and between the analyzed patient groups. *Olsenella*, *Dialister*, *Staphylococcus*, *Campylobacter*, and *Actinomyces* bacteria in samples from WFS patients were found to be more abundant than those identified in alpha diversity in patients with T1DM and in healthy subjects. Interestingly, the bacterial genera found as most common in the present study were also described as predominant in the saliva of pediatric patients with type 1 diabetes [19]. In addition, other studies have found an association between the presence of oral bacteria and the degree of metabolic control of diabetes as measured by HbA1c [20,21]. Significant positive correlations were also observed between HbA1c values and both phylogenetic alpha- (richness) and beta-diversity (compositional variation) of the oral microbiota in gingival samples from children with type 1 diabetes [21]. It is worth noting that in our research, the study group and the comparison group with T1DM were matched both in relation to the metabolic control of diabetes (HbA1c) and the treatment administered (insulin therapy and levothyroxine), which seems to eliminate the influence of these factors on the results obtained. Many studies highlight the two-way relationship between diabetes and periodontitis, indicating that oral and periodontal health should be promoted as an integral part of diabetes care [22,23]. However, it seems that the pathological mechanisms are more complex. It is worth noting that those bacterial genera that most differentiated WFS patients from other groups studied are not typical for diabetic patients. Some of them are specific for periodontal disease, although such disorders were not found in WFS patients in the present study. Thus, it is worth mentioning that periodontitis is a chronic disease, which precedes gingivitis [12]. It has already been recognized that complex interactions between immune response mediators and the bacterial biofilm lead to the progression of these conditions. Dysbiosis disrupts the host response so that most tissue damage is caused by an uncontrolled increase in local inflammation. It results in an increased flow of nutrient-rich GCF and bleeding, causing anaerobic microorganisms to grow in an oxygen-deprived area. It can promote the growth of bacteria residing in the gingival crevice, such as *Porphyromonas gingivalis*. However, the extent to which biofilm accumulation promotes periodontitis is determined by genetic, epigenetic and environmental factors, leading to specific consequences in individuals [12,24,25,26]. Interestingly, these suggestions of genetic-environmental interactions between *P. gingivalis* bacteria and changes in gene expression have already been confirmed in studies on Alzheimer’s disease [13] and even therapeutic options have been found to reduce the bacterial load in the brain [27,28]. However, in the present study, *P. gingivalis* was not found, while several other bacterial genera, including abundant Gram-negative bacteria, were identified as differentiating WFS patients from other study groups. It is important when considering that both other oral pathogenic bacteria and gut microbiota may also be involved in various types of human neurogenesis [14,29,30]. Then, the present study selected the metabolites in GCF such as lactic acid, benzoic acid, and acetic acid, the intensities of which were significantly higher in WFS patients compared with both T1DM and healthy subjects. To date, lactic acid levels in the blood of some WFS patients were found to be elevated and increased after exercise [31]. This phenomena may be of particular interest considering both the primary and secondary mitochondrial dysfunction present in this syndrome [31,32]. Interestingly, the role of lactic acid derivatives as signaling molecules in the brain with possible neuroprotective effects is increasingly being pointed out [33], as is the possibility of using lactic acid bacteria in an animal model of Alzheimer’s disease [34]. Furthermore, the neuroprotective properties of benzoic and acetic acid derivatives have also prompted preclinical trials of their therapeutic use in various neurodegenerative diseases such as amyotrophic lateral sclerosis (ALS) or multiple sclerosis (MS) [35,36], and even applications in DM-induced and Parkinson’s disease (PD)-related neurodegeneration [37]. It should be noted that the results of metabolic studies can be applied not only in the search for markers of observed disorders, but also as therapeutic strategies.

Our study has several limitations. The size of the study group was determined by the low frequency of Wolfram syndrome in the Caucasian population. It would also have been appropriate to expand the control group to include a larger number of healthy, gender- and age-matched individuals. Moreover, in the prevalence analysis of different types of bacteria, there were data without assigned values that refer to unidentified bacteria, which may be related to the collection of biological material for testing. In addition, to define markers of neurodegeneration progression, it would be possible to conduct studies at several time points. Taking this into account, the results obtained are preliminary.

In conclusion, the genome-wide determination of the oral bacterial microflora of the patients with Wolfram syndrome and a comparison with the results of patients with type 1 diabetes to eliminate the influence of diabetes itself and with healthy subjects, allowed the identification of specific oral bacteria in these patients, which may make it possible in the future to determine their role in modulating the neurodegeneration process. However, further studies are needed to define the role of metabolites selected from the GCF as potential biomarkers or therapeutics targeting disorder suppression.

## 4. Materials and Methods

### 4.1. Patients

The study protocol was approved by the University Bioethics Committee at the Medical University in Lodz, Poland (RNN/191/19/KE). Patients and/or their parents gave written informed consent for participation in the study. 

The study group included 12 patients with genetically confirmed WFS, as previously described [8], while the control group consisted of 17 age- (*p* = 0.09) and sex-matched (*p* = 0.91) healthy individuals. The control group included lean people without a clinical diagnosis of diabetes and other chronic diseases, including neurodegenerative diseases. The comparison group included 29 patients with T1DM diagnosed according to the WHO classification, matched for HbA1c level (*p* = 0.23) [38]. All T1DM patients had a minimum of 2 types of autoantibodies, confirming the autoimmune form of diabetes. All patients in the study (WFS patients) and comparison group (patients with T1DM) were treated with subcutaneous insulin therapy. The most common disease coexisting with T1DM is Hashimoto’s disease, which occurred in 7/29 of our patients. No other comorbidities were found in the comparison group. Furthermore, hypothyroidism was diagnosed in 3/12 patients with WFS. All these patients were treated with levothyroxine.

No patient in the study, reference, or control group was diagnosed with periodontal disease. All were from the Caucasian population. Use of oral antibiotics and hormonal contraceptives in the past 2 months, smoking, and pregnancy were taken as exclusion criteria. Detailed characteristics of the individuals studied are shown in Table 2.

In patients in the study, comparison (with T1DM), and control groups, samples of fluid from the buccal mucosa and buccal gingival margin of the first lower permanent molar were collected non-invasively during a routine intraoral dental examination performed by two experienced dentists. It was preceded by correct tooth brushing according to the instructions received. Each patient brushed their teeth for 2 min with an Oral B Genius 9000 electric toothbrush with pulsating and oscillating-rotating motions, equipped with a pressure sensor, in Daily Clean mode with disposable Cross Action tips. It was connected via Bluetooth to the Oral B app for checking the brushing process.

Samples for molecular analysis were collected into sterile screw-cap vials and then stored at −20 °C until further analysis. GCF samples for metabolomic analysis were obtained using sterile PERIOPAPER^TM^ absorbent strips (Oraflow Inc., New York, NY, USA), which were inserted for 30 s into the bottom of the periodontal pocket. Each strip was then placed in an Eppendorf tube with 2% formic acid and frozen at −80 °C for further analysis.

### 4.2. Molecular Analysis—DNA Isolation

Bacterial DNA was extracted from frozen gingival and buccal fluid samples using the Maxwell^®^ RSC Cultured Cells DNA Kit (Promega, catalogue number: AS1620, Madison, WI, USA). 

### 4.3. Library Preparation and Sequencing

Microbial community profiles were assessed by sequencing the 16S rRNA gene. The first step was the amplification of V3 and V4 variable fragments of analyzed gene according to the protocol recommended by Illumina (San Diego, CA, USA). Primers with overhanging adapters compatible with Illumina indexes and sequencing adapters in paired-end sequencing technique were used. Kappa HiFi polymerase (Roche, Mannheim, Germany) was used to amplify fragment with an average 464 bp length. Next the specificity of obtained products was evaluated in an agarose gel and then purified on AMPure XP magnetic beads (Beckman Coulter, CA, USA). The indexing reactions were also carried out using the Kappa HiFi polymerase (Roche, Mannheim, Germany) with the Nextera XT dual-index set (Illumina, San Diego, CA, USA). The concentration of the obtained libraries was determined using the Qubit 2.0 device (Thermo Fisher Scientific, Waltham, MA, USA) and pooled in equal concentrations. The library thus prepared was sequenced on the Miseq platform (Illumina, San Diego, CA, USA) using the kit (MiSeq Reagent Kit v3, 600 cycles).

### 4.4. NGS Data Processing

Obtained raw sequencing data (fastq files) from the Miseq device were uploaded to the Galaxy web platform [39] and we used the public server at usegalaxy.org to analyse the data. FASTQ format files were unified to Sanger FASTQ encoding with FASTQ Groomer tool [40]. Paired end reads were first merged using FLASH tool [41], then the Trimmomatic algorithm was used to remove adapters and low-quality reads (below Q20 value) [42]. Operational Taxonomic Units (OTUs) were assigned by the Kraken 2 algorithm with Standard database [43] and next filtered by classification confidence score at 0.05 level. OTU reads counts for each taxonomy level were extracted into tables and percentage abundance of each identified bacterial taxa was calculated. 

### 4.5. Data Analysis—Alpha and Beta Diversity

The compositional diversity of the microbiome in samples was analyzed at the genus level on web Microbiome Analyst platform [44]. The Shannon, Chaol, ACE, Simpson, and Fisher algorithm and the ANOVA test were used to determine the alpha diversity and the statistical significance differences. In the case of beta-diversity, we used the Principle Coordinate Analysis (PCoA) method to visualize data while statistical significance was tested by the PERMANOVA method. 

### 4.6. Metabolite Extraction and Derivatization from the GCF Samples

Extraction of GCF metabolites was performed as described previously [45] with some modification. To the samples were added 30 μL of O-methoxyamine hydrochloride (20 mg/mL) in pyridine. The vials were closed and vortexed vigorously for 2 min, ultrasonicated for 5 min, and vortexed for 2 min. The vials were covered with aluminium foil and incubated under darkness at room temperature for 16 h. After this time to each sample was added 30 μL of BSTFA with 1% TMCS. The vials were closed and vortexed for 2 min. The GC vials were placed into an oven for 1 h at 70 °C for silylation. Then the samples were cooled for about 1 h at room temperature in the dark. A volume of 60 µL of heptane containing 5 ppm of methyl stearate (MS) was added to each sample and vortexed for 2 min. Quality control (QC) samples were independently prepared by pooling equal volumes of each sample and following the same extraction procedure applied to the experimental samples. An analyte-free extraction blank and reagent blank were also processed. 

### 4.7. Untargeted GC–MS (Gas Chromatography–Mass Spectrometry) Data Analysis

Metabolic fingerprinting was performed using a GC system (series 7890B) equipped with a 7693A auto-sampler and a Mass Selective Detector 7000D (Agilent Technologies, Palo Alto, CA, USA). A volume of 1 µL of the derivatized sample with ISs was injected into a DB–5MS capillary GC column (30 m × 0.25 mm × 0.25 µm) using helium as a carrier gas at a constant gas flow of 1.0 mL/min. The injector temperature was set to 250 °C and the split ratio to 1:10. The temperature gradient program started at 60 °C and was held for 1 min, followed by a subsequent increase in temperature to 320 °C at a rate of 10 °C/min. The GC–MS transfer line, filament source, and quadrupole temperature were set to 280, 230, and 150 °C, respectively. The electron ionization source was set to 70 eV, and the mass spectrometer was operated in the full scan mode, applying a mass range from *m*/*z* 50 to 600 at a scan rate of 1.38 scan/s.

### 4.8. Raw GC–MS Data Processing

The deconvolution and identification were performed using Mass Hunter Quantitative Unknowns Analysis software (B.07.00, Agilent, Santa Clara, CA, USA), alignment with Mass Profiler Professional software (version 13.0, Agilent, Santa Clara, CA, USA), and peak integration using Mass Hunter Quantitative Analysis software (version B.07.00, Agilent, Santa Clara, CA, USA). The identification was performed mainly based on the accurate mass and product ion spectrum matching using the in-house library of authentic standards as well as Fiehn’s and NIST 14 libraries. 

In order to perform the differential analysis of the metabolomics data, the variables were then filtered as described by Godzien et al. [46]. Missing values were replaced by k–means nearest neighbour [47] using the in-house built scripts for MATLAB 7.10 R2010a (MathWorks Inc., Natick, MA, USA).

Before the statistical analysis, clinical sample areas were normalized by IS abundance to minimize the response variability coming from the instrument. Finally, data were filtered based on the coefficient of signal variation (CV) in QC samples, considering values lower than 30% as acceptable.

### 4.9. Visualization of Data and Statistical Analysis

Hierarchical Clustering and Heatmap for molecular analysis were performed using Ward clustering algorithm and Euclidean distance method. Changes in raw abundances between groups were presented using stacked-bar plots. A parametric test ANOVA was used to demonstrate overall changes in the relative abundance of genera. Two-tailed *p* values lower than 0.05 were deemed statistically significant. Above analysis was done on web Microbiome Analyst platform [44]. Categorical variables were presented as numbers with corresponding percentages and means with standard deviations (SD), and continuous variables as medians with interquartile range (25–75%). 

Statistical analyses of metabolites were performed using MetaboAnalyst 5.0 software. Non-parametric ANOVA Kruskal–Wallis test was used to determine if the metabolites were statistically different between the three groups: WFS, T1DM, and healthy participants (*p* < 0.05). When significance was observed, a post-hoc non-parametric Conover-Iman test was applied for pairwise analyses (*p* < 0.05). This test was performed using the R software environment (version 4.0.0, https://www.R-project.org/, accessed on 8 January 2022). In addition, receiver operating characteristic (ROC) analysis was conducted for statistically significant metabolites.

## Figures and Tables

**Figure 1 ijms-24-05596-f001:**
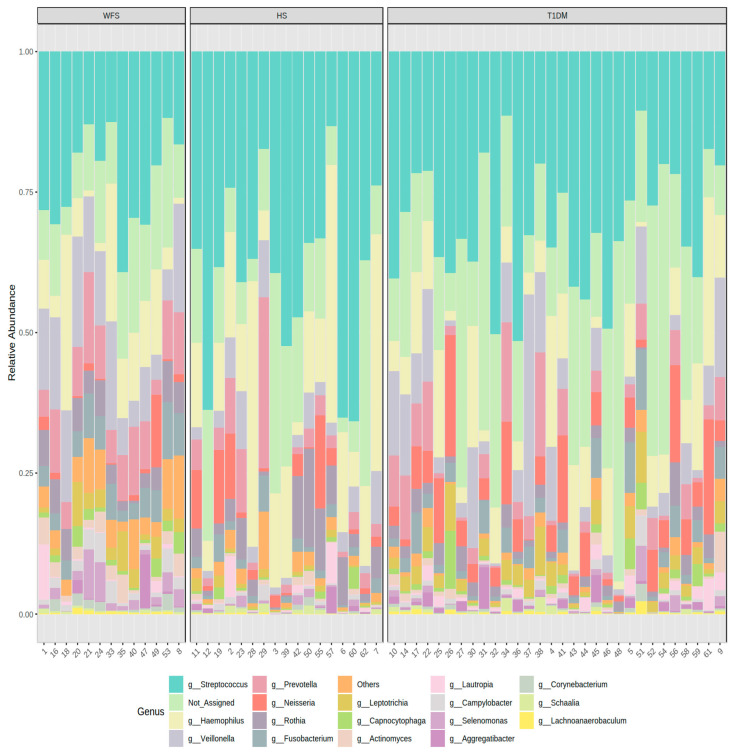
Taxonomic composition of community at bacteria genus level. WFS—Wolfram syndrome; HS—healthy subjects; T1DM—Type 1 diabetes mellitus.

**Figure 2 ijms-24-05596-f002:**
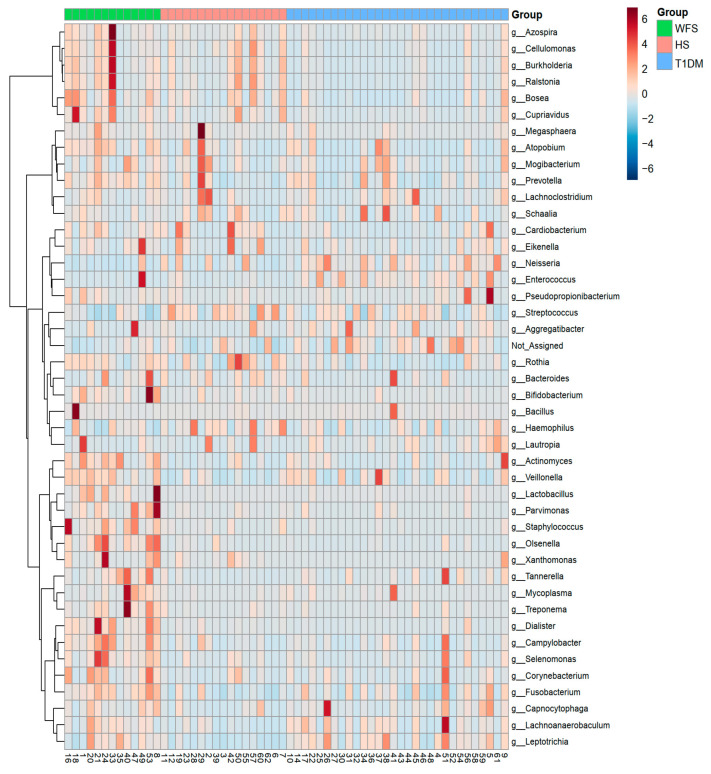
The result of the hierarchical clustering analysis presented as a heatmap. WFS—Wolfram syndrome; HS—healthy subjects; T1DM—Type 1 diabetes mellitus.

**Figure 3 ijms-24-05596-f003:**
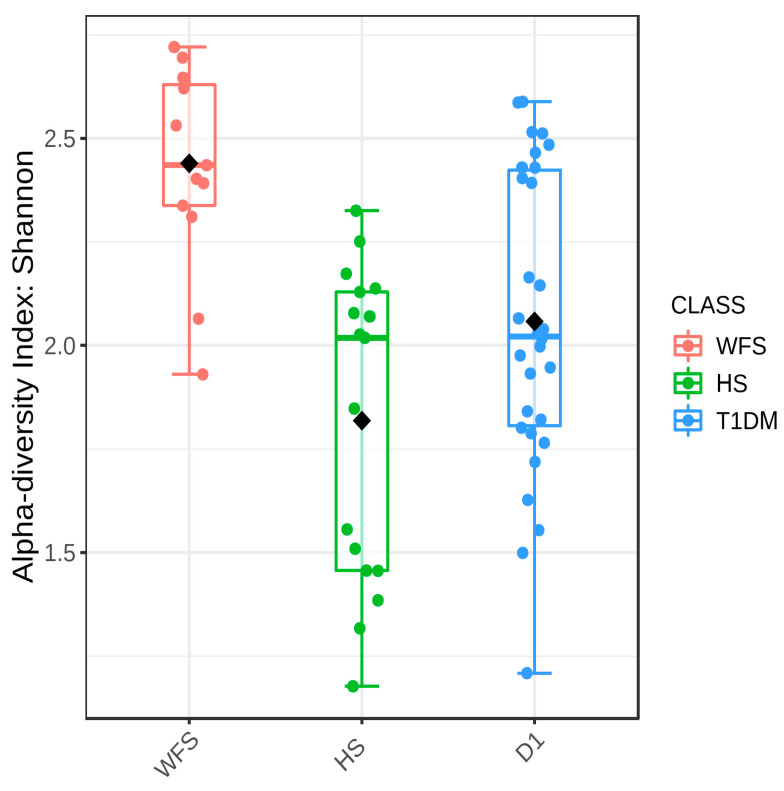
Boxplot showing the overall measure of alpha-diversity in the groups studied using the Shannon method at the bacterial genus level. WFS—Wolfram syndrome; HS—healthy subjects; T1DM—Type 1 diabetes mellitus. The black dot indicates the average value. Statistical significance was evaluated by ANOVA F-value: 12.038; *p*-value < 0.0001.

**Figure 4 ijms-24-05596-f004:**
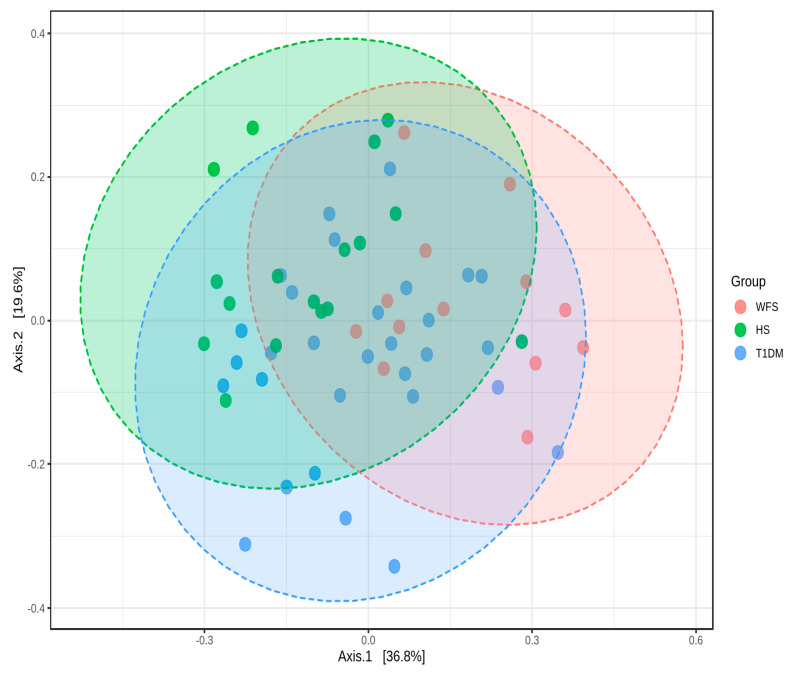
Two-Dimensional Principal coordinates analysis plot using bray distance. Statistical significance was evaluated by PERMANOVA F-value: 6.3631; *p*-value < 0.001.

**Figure 5 ijms-24-05596-f005:**
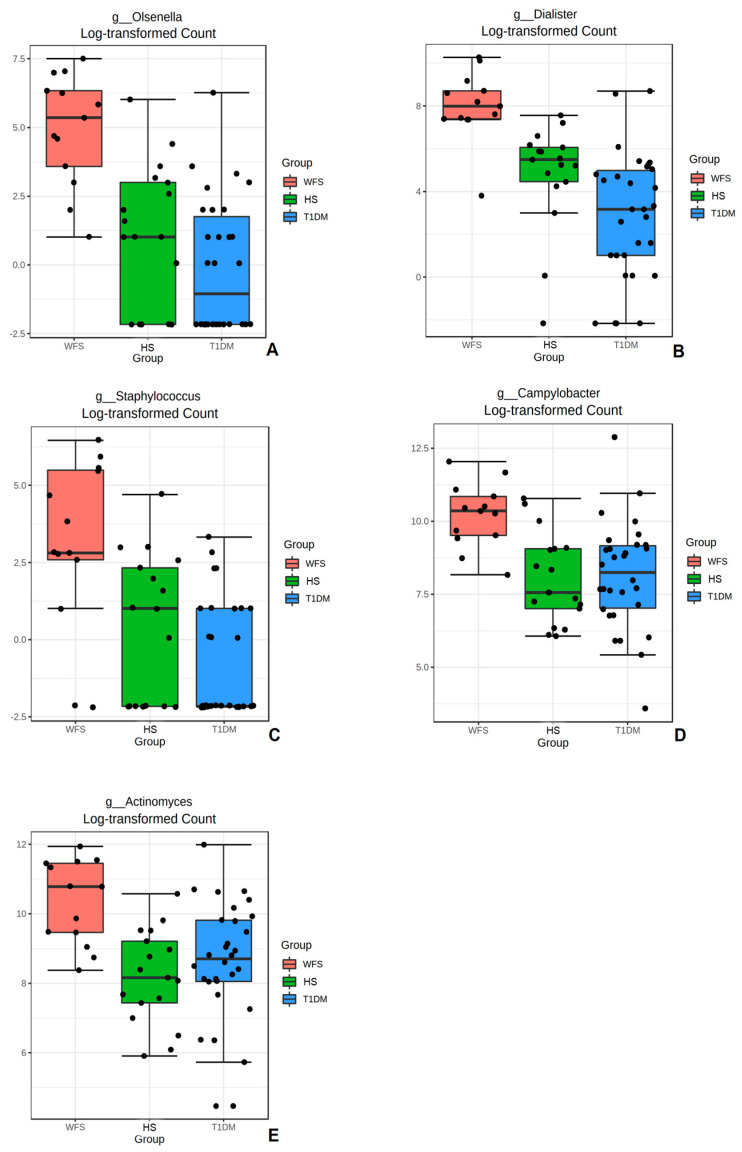
Boxplots comparing abundance for the groups studied for selected bacterial genera: (**A**). Olsenella; (**B**). Dialister; (**C**). Staphylococcus; (**D**). Campylobacter; and (**E**). Actinomyces. WFS—Wolfram syndrome; HS—healthy subjects; T1DM—Type 1 diabetes mellitus. Significant differences in abundance of bacteria genera among the three groups were identified using ANOVA; *p* < 0.001 in all presented bacteria.

**Figure 6 ijms-24-05596-f006:**
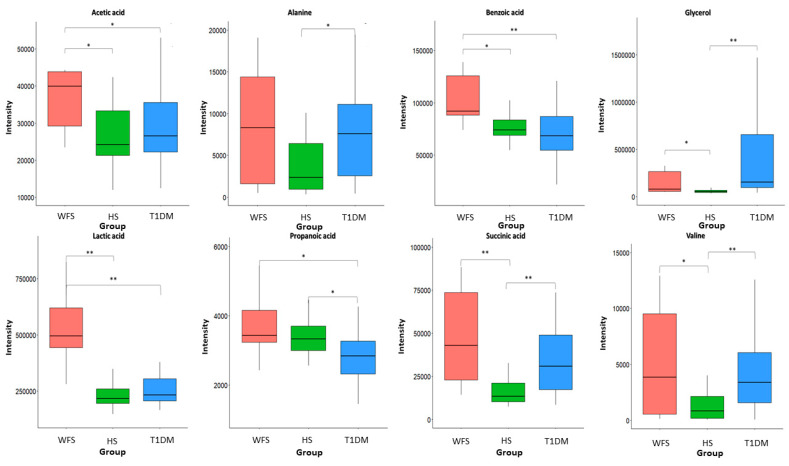
Levels of metabolites discriminating GCF samples of patients with WFS, T1DM, and healthy controls. Significant differences in metabolite intensity among the three groups were identified using a non-parametric Kruskal–Wallis ANOVA (*p* < 0.05), followed by a Conover–Iman post-hoc test (*p* < 0.05 *, <0.01 **). WFS—Wolfram syndrome; HS—healthy subjects; T1DM—Type 1 diabetes mellitus.

**Figure 7 ijms-24-05596-f007:**
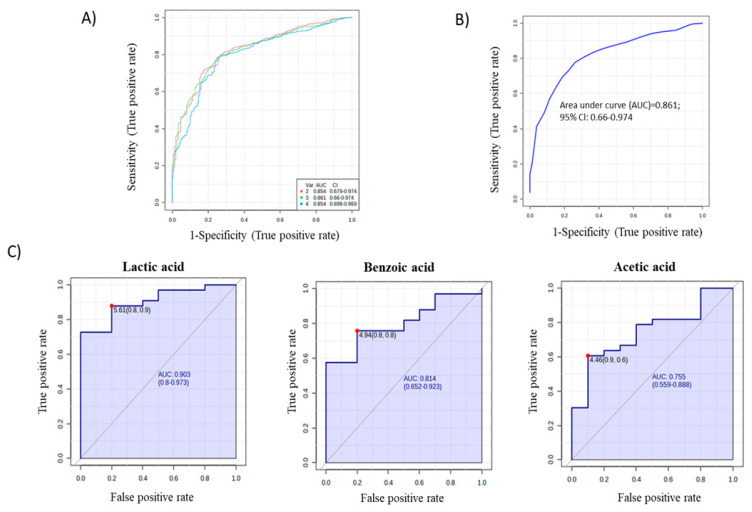
ROC curves and AUC values for all statistically significant metabolites (**A**), multivariate analysis of the ROC curve for the 3 metabolites (**B**), and individual ROC curves for the 3 metabolites (**C**), which can distinguish WFS patients from both T1DM patients and healthy subjects.

**Table 1 ijms-24-05596-t001:** List of metabolites included in the statistical analysis.

Metabolites	RT Library	RT	RI	TI	QI 1	QII 2	CV (%) QCs	HMDB	Group of Metabolites
Acetic acid	-	5.8	596	75	117	45	25.4%	HMDB00237	Carboxylic acids
Propanoic acid	-	6.5	720	131	75	73	16.0%	HMDB00042	Carboxylic acids
Lactic acid	6.851	6.6	732	117	147	73	17.2%	HMDB00190	Alpha hydroxy acids and derivatives
Glycolic acid	7.049	6.9	745	147	73	66	21.2%	HMDB00115	Alpha hydroxy acids and derivatives
Alanine	7.474	7.3	774	116	73	147	27.2%	HMDB00161	Amino acids, peptides, and analogues
Acetic acid	-	7.48	785	145	104	174	23.9%	HMDB00532	Amino acids, peptides, and analogues
Valine	9.151	8.9	898	144	218	73	13.2%	HMDB00883	Amino acids, peptides, and analogues
Glycerol-3-phophate	9.7	9.5	930	299	73	314	12.5%	HMDB00126	Glycerophosphates
Benzoic acid	9.595	9.5	935	179	105	135	19.5%	HMDB01870	Benzoic acids and derivatives
Glycerol	9.941	9.8	950	205	147	73	29.7%	HMDB00131	Carbohydrates and carbohydrate conjugates
Glycine	10.456	10.1	985	174	248	147	24.6%	HMDB00123	Organic acids and derivatives
Succinic acid	10.509	10.4	995	247	73	75	17.9%	HMDB00254	Dicarboxylic acids and derivatives
m-toluic acid	11.006	10.835	1020	193	119	149	17.7%	HMDB62810	Benzoic acids and derivatives

CV, coefficient of signal variation, RT, retention time; HMDB—Human Metabolome Database; TI, target ion; QC, quality control; QI 1, first qualifier ion; QI 2, second qualifier ion; RI, retention index. All compounds reported in the table are three methyls silylated (TMS).

**Table 2 ijms-24-05596-t002:** Clinical characteristics of matched patients with WFS, T1DM, and healthy subjects.

	WFS		T1DM		HS	
	N	Mean ± SD or %	N	Mean ± SD or %	N	Mean ± SD or %
Age (years)	12	23.5 ± 6.2	29	11.3 ± 3.4	17	26.8 ± 3.3
HbA1c (%)	12	7.6 ± 0.6	29	7.3 ± 0.8	17	N/A
Diabetes duration (years)	12	17.9 ± 6.5	29	5.0 ± 2.7	17	N/A
Gender (F/M)	12	8/4 (66.7%/33.3%)	29	14/15 (48.3%/51.7%)	17	11/6 (64.7%/35.3%)

WFS—Wolfram syndrome, T1DM—Type 1 diabetes mellitus, HS—healthy subjects; SD—standard deviation, N/A—not applicable.

## Data Availability

The datasets generated during and/or analyzed during the current study are available from the corresponding author upon reasonable request.

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
