# Peer review of "Evaluation of the Oral Bacterial Genome and Metabolites in Patients with Wolfram Syndrome"

_ijms, 2023, doi:10.3390/ijms24065596_

Round 1
Reviewer 1 Report
In this very interesting paper, the authors have examined the oral microbiome and metabolome of patients with Wolfram syndrome, to provide potential predictors of disease.
Results show several outcomes that clearly differentiate the Wolfram patients from healthy controls and from patients with diabetes, an important control for the Wolfram population.
This paper was well written and the conclusions are well justified; however, I have a few questions for the authors.
1) Were the patients with Wolfram syndrome and the TID patients taking any medications other than insulin?
2) Were the participants' gingival and buccal mucosa samples taken while fasting?
2) The lactic acid metabolomic data were perhaps the most robust and performed the best in the ROC curves. Did the authors examine this outcome correlated to disease progression in the WS patients?
Author Response
Comments and Suggestions for Authors
In this very interesting paper, the authors have examined the oral microbiome and metabolome of patients with Wolfram syndrome, to provide potential predictors of disease.
Results show several outcomes that clearly differentiate the Wolfram patients from healthy controls and from patients with diabetes, an important control for the Wolfram population.
This paper was well written and the conclusions are well justified; however, I have a few questions for the authors.
- Were the patients with Wolfram syndrome and the TID patients taking any medications other than insulin?
R: The Hashimoto disease was present in 7/29 of our patients with T1D. In addition, hypothyroidism was diagnosed in 3/12 of patients with WFS. All of these patients were treated with levothyroxine. 50% of patients with WFS were treated with desmopressin for diabetes insipidus.
- Were the participants' gingival and buccal mucosa samples taken while fasting?
R: Samples of participants' gingival and buccal mucosa were taken after a minimum 2-hour break from eating and after careful brushing of the teeth, according to the instructions provided.
- The lactic acid metabolomic data were perhaps the most robust and performed the best in the ROC curves. Did the authors examine this outcome correlated to disease progression in the WS patients?
R: Thank you for this comment. We fully agree that lactic acid metabolomic data performed the best in the ROC curves. Correlation with disease progression has not been studied, although it would seem interesting to correlate at least lactic acid levels with neuroimaging findings in patients with WFS.
Reviewer 2 Report
The manuscript by Zmysłowska-Polakowska et al. analyzes the oral microbiome and metabolome in WFS patients compared to patients with T1DM and controls. In general, the study is well conducted and has an impact on the neurodegeneration field. However, I have certain concerns (mostly methodological) that need to be cleared up and executed for it to be published on IJMS:
Major concerns
Abstract:
• It is necessary to add background and justify the study of WFS and its approach to neurodegeneration.
Introduction:
• Clarify the characteristics of neurodegeneration present in WFS.
• Highlight the importance or justification of studying WFS.
• Briefly describe the importance of using GWAS in Alzheimer's disease (Analyze and cite the following recent article: doi: 10.3390/ijms24043754)
Methods:
• Given that one of the diagnostic criteria for DM is HbA1c, it is necessary to know the levels in healthy subjects (controls).
• According to the American Diabetes Association (ADA), to make the diagnosis of T1DM, not only the determination of HbA1c is required, but it is also necessary to determine multiple islet autoantibodies (https://doi.org/10.2337/dc22-S002). Please include it in your study.
• Justify why the T1DM group, an autoimmune condition, was used and compare it with the group of patients with WFS, a genetic disorder. Therefore, the pathophysiological mechanisms differ.
• The inclusion and exclusion criteria for healthy subjects are not clear.
• It remains to mention how long they had been on insulin therapy and whether this factor influences the results.
• From a statistical point of view, it is not valid to compare groups that are so unbalanced in their population (number of subjects evaluated per group). Please check and correct.
• Include the demographic characteristics of the patients studied.
• Patients with T1DM may present other associated autoimmune conditions, it is important to specify if the patients included in this study presented them and whether this factor influences the results.
• It is important that the time on insulin therapy of patients with T1DM and WFS is described and whether this may bias the results.
Results:
• The authors comment that they carried out 5 analyzes to determine the alpha diversity analysis, however, they only show one of them in the results. The rest show it as supplementary.
• Please show the results derived from the Analysis of beta diversity.
References:
• 43% of your references are before 2016. Please update them.
Minor concerns:
• In the L40 specify that it is T1DM
• Check the wording of the following sentence: L335-L338.
• At the foot of the figure, mention the n of patients and the statistical test used.
• Improve the quality of Figures 5 and 6.
• Change the term control to healthy subjects.
Author Response
The manuscript by Zmysłowska-Polakowska et al. analyzes the oral microbiome and metabolome in WFS patients compared to patients with T1DM and controls. In general, the study is well conducted and has an impact on the neurodegeneration field. However, I have certain concerns (mostly methodological) that need to be cleared up and executed for it to be published on IJMS:
Major concerns
Abstract:
- It is necessary to add background and justify the study of WFS and its approach to neurodegeneration.
R: We have corrected it.
Introduction:
- Clarify the characteristics of neurodegeneration present in WFS.
R: This was completed.
- Highlight the importance or justification of studying WFS.
R: We have added it.
- Briefly describe the importance of using GWAS in Alzheimer's disease (Analyze and cite the following recent article: doi: 10.3390/ijms24043754)
R: We have added it.
Methods:
- Given that one of the diagnostic criteria for DM is HbA1c, it is necessary to know the levels in healthy subjects (controls).
R: The HbA1c level is only one of several diagnostic criteria by which diabetes can be diagnosed. And it is subject to a number of restrictions, such as its evaluation must be carried out in certified laboratories, using selected methods of determination. In Poland, so far the level of HbA1c has not been allowed as a criterion for diagnosing diabetes. Hence, HbA1c level was not determined in healthy, lean, asymptomatic individuals, without a diagnosis of diabetes.
- According to the American Diabetes Association (ADA), to make the diagnosis of T1DM, not only the determination of HbA1c is required, but it is also necessary to determine multiple islet autoantibodies (https://doi.org/10.2337/dc22-S002). Please include it in your study.
R: In all of these patients, diabetes was diagnosed according to WHO criteria, which, of course, include evaluations of several types of antibodies. In all patients with type 1 diabetes representing the comparison group for WFS patients, a minimum of 2 types of autoantibodies were found, which confirmed the autoimmune form of diabetes in these patients. We have added this information to the manuscript.
- Justify why the T1DM group, an autoimmune condition, was used and compare it with the group of patients with WFS, a genetic disorder. Therefore, the pathophysiological mechanisms differ.
R: In Wolfram syndrome, the first and only symptom preceding the development of neurodegeneration is non-autoimmune insulin-dependent diabetes mellitus, clinically very similar to type 1 diabetes, but without the presence of antibodies in the patients' serum. Hence, type 1 diabetes seems to be the best reference group for evaluating the WFS patients with insulin-dependent diabetes and coexisting neurodegeneration, in order to eliminate the influence of insulin-dependent diabetes itself and look for markers of neurodegeneration alone.
- The inclusion and exclusion criteria for healthy subjects are not clear.
R: We have added it.
- It remains to mention how long they had been on insulin therapy and whether this factor influences the results.
R: Both patients with Wolfram syndrome and selected patients with type 1 diabetes have at least several years of diabetes, so both groups also have a long-term course of insulin therapy (see, please, Table 2).
- From a statistical point of view, it is not valid to compare groups that are so unbalanced in their population (number of subjects evaluated per group). Please check and correct.
R: Thank you for this comment. We realize that the best option from a statistical point of view would be to expand the control group to include more healthy, gender- and age-matched individuals. Therefore, we included this as a limitation of the work, and the results obtained are preliminary.
- Include the demographic characteristics of the patients studied.
R: All patients in the 3 groups are Caucasian, with no ethnic or demographic differences. This is supplemented in the Material and methods section.
- Patients with T1DM may present other associated autoimmune conditions, it is important to specify if the patients included in this study presented them and whether this factor influences the results.
R: The most common disease coexisting with type 1 diabetes is Hashimoto disease, which occurred in 7/29 of our patients. No other comorbidities were found in this group. In addition, hypothyroidism was diagnosed in 3/12 patients with WFS. All these patients were treated with levothyroxine.
- It is important that the time on insulin therapy of patients with T1DM and WFS is described and whether this may bias the results.
R: Both patients with Wolfram syndrome and selected patients with type 1 diabetes have a long-term course of diabetes, so both groups also have a long-term course of insulin therapy, as shown in Table 2. Thus, we decided that the best way to eliminate the differences in the impact of metabolic control of diabetes would be to match them in terms of HbA1c values, which was done.
Results:
- The authors comment that they carried out 5 analyzes to determine the alpha diversity analysis, however, they only show one of them in the results. The rest show it as supplementary.
R: We have added it.
- Please show the results derived from the Analysis of beta diversity.
R: We have added it.
References:
- 43% of your references are before 2016. Please update them.
R: We have improved it to the extent that it was possible. Most of the older publications are related to the methodology itself and the lack of other tools. The remaining papers on the essence of Wolfram syndrome are quite unique, which is related to the ultra-rare nature of the syndrome and the small number of people who deal with it.
Minor concerns:
- In the L40 specify that it is T1DM
R: This is not type 1 diabetes, but insulin-dependent non-autoimmune diabetes. We have added it.
- Check the wording of the following sentence: L335-L338.
R: We have corrected it.
- At the foot of the figure, mention the n of patients and the statistical test used.
R: We have corrected it.
- Improve the quality of Figures 5 and 6.
R: We have corrected it.
- Change the term control to healthy subjects.
R: We have corrected it.
Round 2
Reviewer 2 Report
The authors significantly improved their manuscript, however, there are still some details that need to be clarified:
1. It is still not clear how it was determined (biochemically, clinically, or by imaging) that the healthy subjects did not have diabetes or any neurodegenerative disease. Please specify.
2. The following sentence remains to be included and discussed in the manuscript: “The most common disease coexisting with type 1 diabetes is Hashimoto's disease, which occurred in 7/29 of our patients. No other comorbidities were found in this group. Furthermore, hypothyroidism was diagnosed in 3/12 patients with WFS. All these patients were treated with levothyroxine.
Author Response
The authors significantly improved their manuscript, however, there are still some details that need to be clarified:
- It is still not clear how it was determined (biochemically, clinically, or by imaging) that the healthy subjects did not have diabetes or any neurodegenerative disease. Please specify.
R: We have corrected it.
- The following sentence remains to be included and discussed in the manuscript: “The most common disease coexisting with type 1 diabetes is Hashimoto's disease, which occurred in 7/29 of our patients. No other comorbidities were found in this group. Furthermore, hypothyroidism was diagnosed in 3/12 patients with WFS. All these patients were treated with levothyroxine.
R: We have added it and discussed it.